# *Prolucanus beipiaoensis* gen. et sp. nov.: The First Fossil Species of Lucaninae (Coleoptera: Lucanidae) from the Early Cretaceous of Northeastern China

**DOI:** 10.3390/insects13030272

**Published:** 2022-03-10

**Authors:** Zhi-Hao Qi, Erik Tihelka, Chen-Yang Cai, Hai-Tian Song, Hong-Mu Ai

**Affiliations:** 1College of Plant Protection, Fujian Agriculture and Forestry University, Fuzhou 350002, China; qizhihao2021@126.com; 2Fujian Academy of Forestry Sciences, Fuzhou 350012, China; 3State Key Laboratory of Palaeobiology and Stratigraphy, Nanjing Institute of Geology and Palaeontology, Center for Excellence in Life and Paleoenvironment, Chinese Academy of Sciences, Nanjing 210008, China; wn20250@bristol.ac.uk (E.T.); cycai@nigpas.ac.cn (C.-Y.C.); 4School of Earth Sciences, University of Bristol, Life Sciences Building, Tyndall Avenue, Bristol BS8 1TQ, UK

**Keywords:** stag beetle, Lucaninae, fossil, *Prolucanus*, new species, Yixian Formation, China

## Abstract

**Simple Summary:**

The family Lucanidae Latreille, 1804, has long been considered to be one of the most archaic groups in the superfamily Scarabaeoidea and contains more than 110 genera and about 1300 extant species. Most adult lucanids display obvious sexual dimorphism as males are often larger and possess well-developed and variable mandibles used to compete with other males. For a long period of time, researchers of stag beetles have had problems and disputes, because many views are only speculation and cannot be supported by the evidence of fossil specimens. In this paper, we describe and illustrate a new fossil species from the Early Cretaceous of northeastern China: *Prolucanus beipiaoensis* **gen. et sp. nov.** Based on this well-preserved specimen, a new genus is established, representing the first record of the fossil Lucaninae in China and the oldest reliable record of the subfamily.

**Abstract:**

A new stag beetle fossil is described from the Yixian Formation in northeastern China. A new genus is established based on this well-preserved specimen. *Prolucanus beipiaoensis* **gen. et sp. nov.** is firmly placed in the extant lucanid subfamily Lucaninae based on its broad prosternal process and geniculate antennae. Our discovery represents the first fossil record of Lucaninae from the Late Mesozoic of China and the oldest reliable record belonging to the subfamily. We also summarize the fossil species of stag beetles found in China and the described fossil species of the subfamily Lucaninae found in the world. A key to all fossil genera of Lucanidae from China is provided.

## 1. Introduction

The family Lucanidae Latreille, 1804, is long considered to be one of the earliest diverging lineages of the superfamily Scarabaeoidea Latreille, 1802, and contains more than 110 genera and about 1300 extant species distributed throughout all main zoogeographical regions except Antarctica [1,2,3,4,5,6]. Most adult lucanids display obvious sexual dimorphism. Males are often larger and possess well-developed mandibles used to compete with other males, whereas females often possess smaller mandibles, which are more conducive for activities such as oviposition. Crown-group representatives of Lucanidae are divided into four extant subfamilies: Lucaninae, Lampriminae, Syndesinae, and Aesalinae [2]. Four extinct subfamilies (Protolucaninae, Paralucaninae, Ceruchitinae, and Litholampriminae) have been established [7,8], although the validity of some of them has been questioned.

Fewer than 30 species of fossil stag beetles have been described, including two recently discovered Cretaceous species preserved in amber from northern Myanmar [9,10]. To date, six lucanid fossils have been reported in China [8,11,12]. *Juraesalus atavus* Nikolajev et al., 2011, the oldest stag beetle fossil, originates from the Middle Jurassic deposits near Daohugou Village, Ningcheng County, Chifeng City, Inner Mongolia, China, and was placed in the subfamily Aesalinae. *Sinaesalus tenuipes* Nikolajev et al., 2011, *S*. *longipes* Nikolajev et al., 2011, and *S*. *curvipes* Nikolajev et al., 2011 originate from the Lower Cretaceous Yixian Formation in the vicinity of the Yangshuwanzi Village, Ningcheng County, in Inner Mongolia and were placed in the subfamily Aesalinae. *Prosinodendron krelli* Bai, Ren et Yang, 2012 was collected from the Lower Cretaceous Yixian Formation near Chaomidian Village, Beipiao City, Liaoning Province, and placed in the subfamily Syndesinae. *Litholamprima longimana* Nikolajev et Ren, 2015 from the Lower Cretaceous Yixian Formation near Chaomidian Village, Liaoning Province, was assigned to the new fossil subfamily Litholampriminae and erected to accommodate the taxon.

Up until now, only nine fossil species of Lucaninae have been described in the world [4,13]. There is an extinct genus from the Mesozoic, including three species [14]: *Cretolucanus longus* Nikolajev 2007, *C*. *ordinarius* Nikolajev 2007, and *C*. *sibericus* Nikolajev 2007. They were previously considered to be the oldest fossil species of Lucaninae, originating from the Early Cretaceous in Pad Semen, Russia. However, no convincing characters can be found on three very vague fossil specimens to support their classification as Lucaninae (see Discussion). There are five genera (two extinct and three extant) and six species from the Cenozoic: *Succiniplatycerus berendti* (Zang, 1905) originates from the Eocene in Baltic Amber. *Miocenidorcus andancensis* Riou, 1999 originates from the Miocene in Andance, France. *Platycerus sepultus* Germar, 1837 originates from the Oligocene, “incarbone fossili territorii Rheni prope Bonnam”, Germany. *Platycerus zherichini* Nikolajev, 1990 originates from the Oligocene, Pozhar region in Russia. *Dorcus primigenius* Deichmüller, 1881 originates from the Eocene, Kučlín near Bílina in the Czech Republic. *Lucanus fossilis* Wickham, 1913 originates from the Oligocene, Florissant in the USA. In conclusion, the previous reliable record of the oldest fossil Lucaninae was determined to be from the Eocene.

In this paper, we describe and illustrate a new fossil species from the Lower Cretaceous Yixian Formation near the Huangbanjigou Village in Beipiao City, Liaoning Province. Based on this well-preserved specimen, a new genus is established, representing the first record of fossil Lucaninae in China and the oldest reliable record of the subfamily.

## 2. Material and Methods

The present specimen originates from the fossiliferous yellowish tuffs of the Lower Cretaceous Yixian Formation from the locality near the Huangbanjigou Village of Beipiao City in the Liaoning Province of northeastern China. This stratum represents the famous Jehol Biota known for its feathered dinosaurs, mammals, birds, angiosperm plants, and numerous insects [15]. Its geological age has been indicated with respect to the Early Cretaceous, circa 125 MYA [16].

Images were taken by using a Canon 5D mark IV digital camera with a 100 mm f/2.8 macro lens. A Canon MT-26EX twin flash was used as the light source. Morphological details were photographed by using a Keyence VHX-5000 digital microscope with the Keyence VH-Z20R zoom lens (20–200×). The images were processed and combined into figures by using Adobe Photoshop CC 2019. The type specimen is housed at the Nanjing Institute of Geology and Palaeontology (NIGP), Chinese Academy of Sciences, in Nanjing, China.

Measurement criteria in millimeters (mm) are used as follows:
Body length: length between the apex of mandible to the elytral apex along the midline;Elytral length: length between the basal border and the apex of elytra along suture;Elytral width: widest part of both elytra combined;Head length: length between the anterior apex of clypeus and the posterior margin of occiput along the midline;Head width: widest part of head;Mandible length: length from the apex of mandible to its base at anterior margin of the head;Pronotal length: length of the pronotum along the midline;Pronotal width: widest part of pronotum.

## 3. Results


**Systematic Palaeontology.**


Order **Coleoptera** Linnaeus, 1758

Superfamily **Scarabaeoidea** Latreille, 1802

Family **Lucanidae** Latreille, 1804

Subfamily **Lucaninae** Latreille, 1804

Genus ***Prolucanus*** gen. nov.

**Type species:** *Prolucanus beipiaoensis* sp. nov.

**Diagnosis.** The new genus is firmly placed in the subfamily Lucaninae based on broad prosternal process and geniculate antennae, and it can be distinguished from almost all fossil or extant genera by a combination of the following features: (1) body broadly-oval; (2) eyes large, nearly entire, slightly divided by ocular canthus; (3) antenna geniculate, antennal club with four antennomeres; (4) pronotum transverse, nearly trapezoidal, over twice as broad as long; and (5) mesotibia only with one long and sharp spur at apex (maybe only one preserved).

**Description.** Body broadly-oval. Head large, wider than long; preocular angle rounded; eyes large, slightly divided by ocular canthus; antenna geniculate with 10 antennomeres, antennomere I elongate and strong, antennal club with four antennomeres, antennomere VII slender and sharply pointed apically, antennomeres VIII–X lamellate; mandible robust, widest and strongly inward curved near the base, mandibular apex bidentate, incisor edge of mandible smooth, without tooth (female). Pronotum transverse, over twice as broad as long, front angle rounded. Elytra elongate, densely punctate. Legs strong, protibia with 3–4 teeth along outer margin, progressively larger from base to apex, apex bifurcate, short, nearly right angle; mesotibia with one large tooth on outer margin and at least one long and sharp spur at apex. Prosternal and mesosternal process broad. Abdomen five-segmented, apex slightly rounded.

**Etymology.** The Latin term “pro-”, meaning ancient or primitive, and *Lucanus* after the type genus of Lucaninae.


***Prolucanus beipiaoensis* sp. nov.**


(Figure 1, Figure 2 and Figure 3)

**Material examined.** Holotype: NIGP176633, probably female, a well-preserved and almost complete body collected from the Yixian Formation near the Huangbanjigou Village of Beipiao City, Liaoning Province, and is now housed in the Nanjing Institute of Geology and Palaeontology, Chinese Academy of Sciences, Nanjing, China.

**Diagnosis.** As for the genus, vide supra.

**Description.** Probably female (see Section 4). Body broadly oval, length 14.5 mm. Length (mm) of different body parts: head (2.0), mandible (1.0), pronotum (2.7), and elytra (9.1); width: head (3.7), pronotum (6.5), and elytra (7.8).

Head (Figure 2A,B and Figure 3A) transverse, 1.9 times as wide as long, broadest near base, densely punctate. Preocular angle rounded, preocular margin short. Eyes large, nearly entire, slightly divided by ocular canthus. Antenna (Figure 3C) geniculate with 10 antennomeres, antennomere I elongate and strong, antennal club with four antennomeres, antennomere VII slender and sharply pointed apically, antennomeres VIII–X lamellate. Mandibles (Figure 3B) robust, widest and strongly inward curved near base, mandibular apex bidentate, incisor edge of mandible smooth, without tooth. Mentum (Figure 3B) nearly trapezoidal, anterior margin slightly emargination. Gula large.

Pronotum (Figure 2A,B and Figure 3A) transverse, form 2.4 times as wide as long, widest near base. Anterior margin is a relatively straight arc, front angle (Figure 3A) protruding forward and rounded. Lateral margin nearly smooth. Posterior margin longer than anterior margin and protrudes obviously to the rear at the midline.

Scutellum not visible.

Elytra (Figure 2A,B) elongate, densely punctate, wider than pronotum, elytron 2.3 times longer than wide. 

Legs (Figure 2A,B and Figure 3A,D–H) strong. Protibia (Figure 3D) gradually widened apically and with 3–4 teeth along outer margin, gradually larger apically, apex bifurcate, short, nearly right angle. Mesotibia (Figure 3F,G) with one large tooth on outer margin and at least one long and sharp spur at apex. Metacoxa, metafemur and metatibia (Figure 3H) burly.

Ventral side clearly preserved. Procoxal cavities large, strongly transverse, broadly separated by wide prosternal process (Figure 3A). Mesocoxal cavities oval, broadly separated by wide mesosternal process (Figure 3E). Metacoxal cavities large, strongly transverse. Sutura sternalis (Figure 3E) obvious. Abdomen (Figure 3H) five-segmented, first ventrite not completely divided by metacoxae, apex rounded.

**Etymology.** The specific name is derived from the type locality.


**Key to known fossil genera of Lucanidae from China**
Antenna geniculate..............................................................................*Prolucanus* **gen. nov.**
−Antenna partially or not geniculate....................................................................................2
2Head width exceeding more than half of pronotal width...............................................3
−Head not broader than half the width of pronotum........................................................4
3Prosternal process broad.................................................*Juraesaluss* Nikolajev et al., 2011
−Prosternal process narrow...............................*Prosinodendron* Bai, Ren, and Yang, 2012
4Length of scape longer than length of pedicel..............*Sinaesalus* Nikolajev et al., 2011
−Length of scape equal to or shorter than length of pedicel.........................................................................................................................*Litholamprima* Nikolajev and Ren, 2015


## 4. Discussion

Extant stag beetles can be classified relatively accurately into subfamilies by a combination of external and genital morphology and molecular means. The genital morphology of lucanids is an important character [17,18], but it is usually not visible in fossil species and can only be classified by external morphological characters. Holloway [19] considered the canthus as absent in only a small number of Lucanidae, and the eyes are stated to be entire. Lampriminae, Aesalinae, and Syndesinae all have entire eyes, whereas Lucaninae all have varied forms of canthus. However, Howden and Lawrence [20] expressed reservations and proposed Aesalinae to also have eyes partly divided by a canthus, and the tribe Platycerini (Lucaninae) may have the eyes entire; thusm this character should be used with caution. According to Howden and Lawrence [20], the subfamily Lucaninae is characterized by the following combination of characters: (1) prosternal process not or only slightly narrowed between coxae so that the latter are distinctly separated; (2) antenna geniculate; and (3) body elongate and depressed. Although the above characters selected by the authors only cover North American species, these morphological characters are applicable to most Lucaninae (except *Chiasognath**us*, *Ryssonotus*, *Pholidotus*, etc., in which the body is usually oval, convex, and the prosternal process is narrow). In 2007, Holloway [2] regarded the following two characters as the unique characters of Lucaninae in New Zealand: (1) prementum situated near the middle of the mentum and not visible when the head is viewed from below; (2) basal first segment of labial palp concealed by the mentum. However, these two characters also apply to all species of Lucaninae, worldwide. Unfortunately, these parts are difficult to preserve intactly in most fossil specimens and in our fossil specimen. *Prolucanus* gen. nov. can be firmly placed in the extant subfamily Lucaninae based on the combination of morphological characters mentioned above. Since most lucanid adults display obvious sexual dimorphism, with males being larger and possessing prominent mandibles, the fossil specimen with a broadly oval body, short and robust mandible, and possessing only one inner tooth is likely to be female.

Based on three fossil specimens discovered from the Early Cretaceous in Pad Semen, Russia, Nikolajev [14] established the extinct genus *Cretolucanus* Nikolajev, 2007 and described three species (*C*. *longus*, *C*. *ordinarius*, and *C*. *sibericus*), classified them into Lucaninae. However, the description, diagnosis, and illustrations provided by the author are confusing. The author distinguished *Cretolucanus* from members of Penichrolucaninae, Aesalinae, and Nicaginae (merged with Aesalinae) only by its elongated body, small head, and the eyes partly divided by ocular canthus; moreover, unfortunately, the author did not even provide any evidence supporting its subfamilial placement in Lucaninae. In fact, it can be seen from the illustrations that the three fossil specimens are very vague, and it is almost impossible to extract effective characters for comparison. Only *C*. *longus* preserved part of the antennal structure, which can be seen as not geniculate, which is obviously in contradiction with Lucaninae. In addition, there are also insufficient characters to indicate that it belongs to the family Lucanidae. Therefore, we recommend that *Cretolucanus* should be removed from Lucaninae and tentatively designated as Lucanidae subfamilia incerta or Scarabaeoidea familia incerta until more complete specimens are found.

## 5. Conclusions

Lucaninae is the most species-rich and morphologically diverse subfamily of stag beetles. Since there are many genera and species of Lucaninae and convergent evolution is common, it is difficult to clarify the relationship between various genera and tribes by using traditional morphological classification. In 2015, Kim and Farrell [4] reconstructed the first molecular phylogeny of world stag beetles, and a time-calibrated phylogeny based on five fossil data was estimated. They demonstrated that the subfamilies Lucaninae and Lampriminae appeared monophyletic under both methods of phylogenetic inferences; however, Aesalinae and Syndesinae were found to be polyphyletic. Within Lucaninae, the most primitive Platycerini lineage branched off first during the mid-Early Cretaceous around 125 MYA, and the other lineage further diverged into the two main lineages at the end of Lower Cretaceous circa 108 MYA, which then diversified rapidly in each hemisphere. *Prolucanus beipiaoensis* gen. et sp. nov. is basically consistent with the research results of Kim and Farrell [4]. Based on morphological comparison, we believe that the new genus does not belong to the branch of Platycerini lineage because of its broadly oval body, transverse and large head, trapezoidal and transverse pronotum, outer margin of protibia with a few isolated teeth, and abdomen with five visible ventrites. However, Platycerini usually with elongated body, longitudinal and smaller head, nearly oval pronotum, outer margin of protibia serrate continuously, and abdomen with six visible ventrites. In addition, the character of having large eyes that are nearly entire and slightly divided by an ocular canthus as found in *Prolucanus* gen. nov. and Platycerini may be a plesiomorphic feature, which is different from most Lucaninae. Therefore, we suggest that *Prolucanus* gen. nov. is one of the earliest diverging lineages in Lucaninae.

## Figures and Tables

**Figure 1 insects-13-00272-f001:**
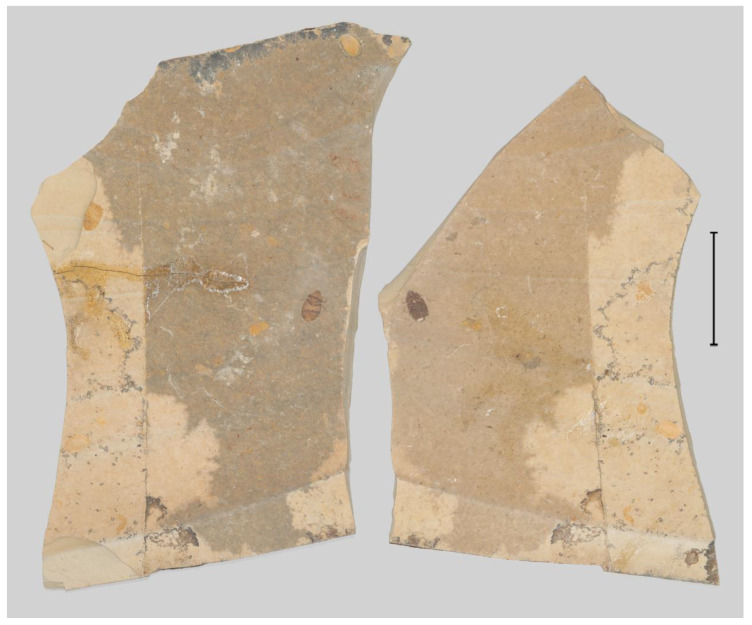
Fossil specimen of *Prolucanus beipiaoensis* gen. et sp. nov. (holotype, NIGP 176633). Scale bars = 5 cm.

**Figure 2 insects-13-00272-f002:**
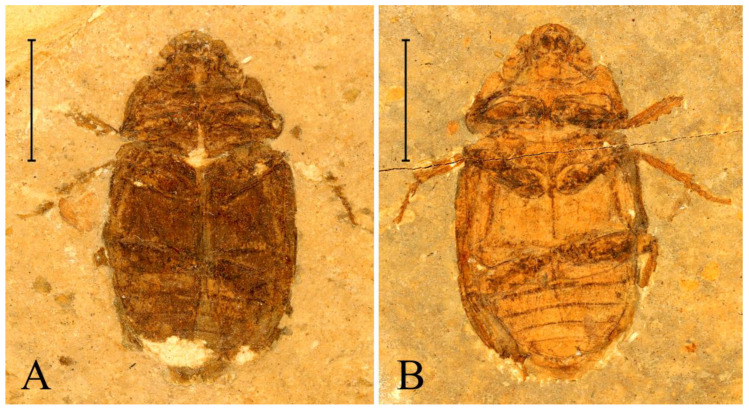
Habitus of *Prolucanus beipiaoensis* gen. et sp. nov. (holotype, NIGP 176633). (**A**) part; (**B**) counterpart. Scale bars = 5 mm.

**Figure 3 insects-13-00272-f003:**
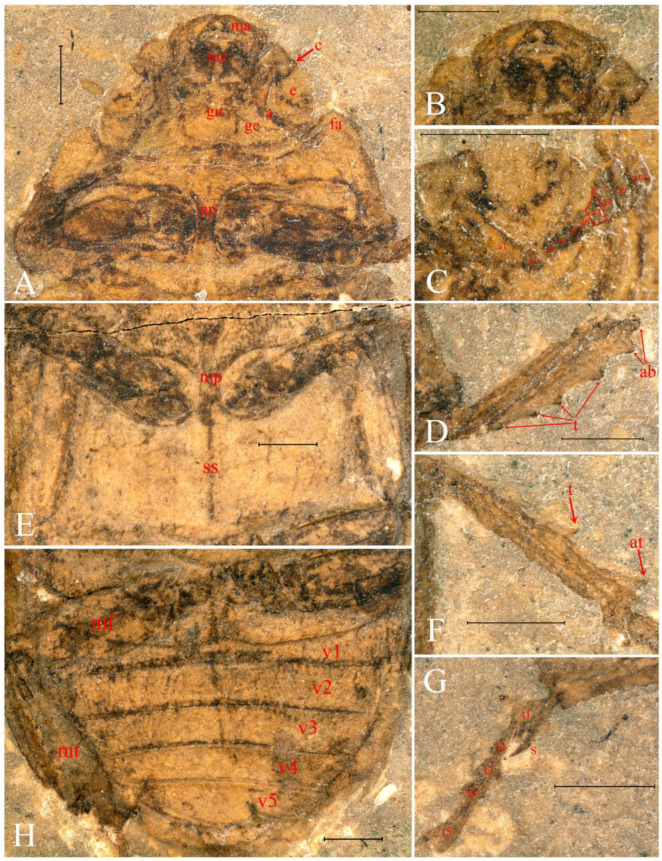
Morphological details of *Prolucanus beipiaoensis* gen. et sp. nov. (holotype, NIGP176633). (**A**) Head ventral and prosternum; (**B**) mandibles and mentum; (**C**) antenna; (**D**) protibia; (**E**) mesosternum and metasternum; (**F**) mesotibia; (**G**) mesotibial apical spur and mesotarsus; (**H**) abdomen and hind leg. Abbreviations: a, antenna; a1–10, antennomere 1–10; ab, apex bifurcate; at, apical tooth; c, canthus; e, eye; fa, front angle; ge, gena; gu, gula; ma, mandible; me, mentum; mf, metafemur; mp, mesosternal process; mt, metatibia; pp, prosternal process; s, spur; ss, sutura sternalis; t, tooth; t1–5, tarsomeres 1–5; v1–5, ventrite 1–5. Scale bars = 1 mm.

## Data Availability

The study did not report any data.

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
