# Peer review of "Prolucanus beipiaoensis gen. et sp. nov.: The First Fossil Species of Lucaninae (Coleoptera: Lucanidae) from the Early Cretaceous of Northeastern China"

_insects, 2022, doi:10.3390/insects13030272_

Round 1
Reviewer 1 Report
The fossils are definitely worth being published. However, the figures are not of optimal quality and need to be improved (see below). Also in the text, a larger number of grammar and other language errors need to be corrected (also see below). The authors state that other studies do not clearly show their results, but then it is necessary to be as precise as possible in the own study.
In summary, I recommend publication after minor revision.
General: I always recommend to omit Linnean ranks, especially those above genus level (family, subfamily, etc.). Though widely used, they have in fact no scientific meaning as there are no objective criteria when a group is a family, when a subfamily, and so on. In most cases, it is rather easy to phrase the sentences slightly different to work also without the Linnean rank.
Simple summary: In my view, a simple summary means that any entomologically interested person can read it without having to look up any special terms. However, in this summary, several group names occur (e.g., Lucanidae, Platycerini), which need to be explained. Please think more of the reader. In fact, the simple summary reads more like an abstract to me, and as the abstract is rather short, the authors might consider to used the simple summary as abstract and write a new simple summary instead.
20: After “sp. nov.” I would end the sentence and start a new one with “Based on”.
21: I recommend the construction “fossil representatives of Lucaninae” as Lucaninae is a group name and includes also extant representatives. Also check in remaining text.
21: reliable instead of believable
33: instead of “basalmost”, which strongly depends on the way the phylogeny is presented, I recommend “earliest offshoots within Scarabaeoidea”
38/39: Crown-group representatives of Lucanidae
40: four with small initial
47: If one talks about species, I recommend to write “species” instead of “taxa”. Check also in remaining text.
51: “placed in” always sounds like a decision by the author, but the process in fact means that the characters point to a position within a certain group. I recommend to rephrase it here and in other places in the text.
54: “the Aeselinae”: I was told by a philosophically educated biologist that there should be no “the” in front of group names as the group is an individual or natural kind; “the” would only be placed in front of logical classes. Please also check in the remaining text.
61: from the Mesozoic
63: originating
64: unfounded instead of groundless
65: Is specimens correct? So 3 species erected on only 1 specimen each?
67: from the Cenozoic
67: I assume it should be (Zang, 1905)
69/70: “incarbone fossili territorii Rheni prope Bonnam” needs to be translated; I presume it means something like “coalified/carbonised fossil in the Rhine region near Bonn”, but please ask someone who really can translate it
72: Here and in the other cases, I recommend to check what the current writing is. Especially for areas surrounding Germany, there has been a totally different way to write these city names, and several of them might be considered offensive today as they go back to the Third Reich. It is not Kutschlin near Bilin anymore, but Kučlín near Bílina.
74: reliable instead of believable; also check in remaining text
74: Please add Lucaninae
83: I recommend some systematic sorting of the groups.
112/114: Could you please provide the vernacular names also in English?
121: “canthus slightly not obvious divide of eyes” does not sound like a proper sentence
121: Antenna with capital initial
124: “curved near” (delete “at”)
125: smooth instead of smoothness
131: pro in Latin means before, maybe also ancient, but I see no reason for primitive
140: Body with capital initial
148: “curved near” (delete “at”); this “at near” construction occurs also at other places, please check and change
165: Sternuma sulko seems a very special term to me not used in all insects, better explain it.
165: obvious instead of obviously
182: not broader instead of no broader
189ff: That is a very long sentence, please make two out of it.
192: considers instead of consider
204: “It is worth mentioning...”, this sentence is very incomplete, please rephrase.
208: not in fossil species, but in fossil specimens
213: originating instead of originate
217: distinguishes instead of distinguish
217: What is meant here with “it”? Be more precise.
219: Again, what does “it” mean here?
219: does not instead of doesn't
225: is removed instead of be removed
232: “based on five fossil data estimated”, this sentence makes no sense to me
232: They instead of The
235: “primitive” has a certain bad “smell”, I recommend to rephrase “the lineage of Platycerini with the most ancestral characters”
240: it has a (add a)
242: an elongated body (add an); also add a/an in remaining sentence
244: what is meant with “them”?
240–244: These sentences have several grammar errors, as I partly mentioned in the previous comments. Please read carefully and correct.
298: Syndesus and Sinodendron need to be in italics.
Both figures would benefit from optimisation of the histogram, which would increase the contrast. Fig. 2 is additionally very greyish, looks like only a small part of the entire brightness range is used (also here histogram optimisation helps).
The letters in Fig. 2 are of very different size, and the smallest ones are really very small, hard to read. Try to make them a bit more uniform within that figure.
Furthermore, I think that black or white labellings in the figures will work, if the authors use a thin stroke around the letters. The red colour is rather disturbing and still does not provide a good contrast.
Author Response
Dear Reviewer,
Thank you very much for your valuable comments on the manuscript. We have revised and improved most of your suggestions accordingly.
Yours sincerely,
Author
Reviewer 2 Report
the paper is interesting with good description and illustrations
it should be essentially improved:
- it is desirable to note author and year for subfamilies and tribes when they are first mentioned;
- it is necessary to include comparison with other lucanid subfamilies (including fossil groups even unknown to the authors) with compsrison of characters listed in the description and and in the description of ;
- the Discussion in the present version has three paragraphs: 1 - literature review of classifications of modern fauna; 2 - explanations absence of comparision with fossil lucanids - it would be desirable to analyse the available characters (at least shortly); 3 paragraph is devoted a phylogenetic concept by the authors - it should be smoothed and demonstrate that some correspondence of characters can be interpreted by a certain hypothesis taking into consideration that this hypothesis is still not proved by great number of facts;
- other recommendation are in the file attached

Author Response
Dear Reviewer,
Thank you very much for your valuable comments on the manuscript. We have revised and improved most of your suggestions accordingly.
However, with regard to "comparison with other lucanid subfamilies", the new genus is firmly placed in the existing lucanid subfamily, Lampriminae family based on the combination of morphological characteristics. So we think it's not necessary to add comparisons between subfamilies. Comparisons with other fossil species can be distinguished by the combination of diagnostic features. In addition, some of the described stag beetle fossils in the world are incomplete and cannot be used as a key point of comparison.
Kind regards,
The authors
Reviewer 3 Report
In my opinion the manuscript is well prepared. There are only some places which should be corrected before the final publication. These latter are marked in the attached file and specified below.
Moreover, I have some substantional remarks which should be considered before the publicstion.
1) The genus should be compared with some similar taxa (it is not enough to write that it is unique, please specify the features and compare them with specific taxa).
2) Similarly, the issue of belonging to a subfamily should be indicated in the introduction or in the diagnosis of the genus, not in the summary.
3) The key should also be for all fossil Lucaninae / Lucanidae and not only from China (?)

Author Response
Dear Reviewer,
Thank you very much for your valuable comments on the manuscript. We have revised and improved most of your suggestions accordingly.
However, "The genus should be compared with some similar taxa" is difficult to achieve, because convergent evolution is common in stag beetles. Many tribes, genera, or species have similar appearances, they are mainly characterized by reproductive features that are not possible in fossils. In addition, some of the described stag beetle fossils in the world are incomplete and cannot be used as a key point of comparison.
Kind regards,
The authors